# Roasted Wheat Germ: A Natural Plant Product in Development of Nutritious Milk Pudding; Physicochemical and Nutritional Properties

**DOI:** 10.3390/foods11121815

**Published:** 2022-06-20

**Authors:** Mahsa Majzoobi, Fatemeh Ghiasi, Mohammad Hadi Eskandari, Asgar Farahnaky

**Affiliations:** 1Biosciences and Food Technology, School of Science, RMIT University, Bundoora Campus, Melbourne, VIC 3083, Australia; asgar.farahnaky@rmit.edu.au; 2Department of Food Science and Technology, School of Agriculture, Shiraz University, Shiraz 7144165186, Iran; fatemeh.ghiasi@gmail.com (F.G.); eskandari@shirazu.ac.ir (M.H.E.)

**Keywords:** healthy foods, milk pudding; plant protein, wheat germ; nutritional value

## Abstract

Wheat germ has been recognized as an economical source of high-quality plant proteins and bioactive compounds for food fortification. Thus, it can be used for valorization of food products as a feasible strategy to enhance the nutritional quality and reduce wheat milling waste. In this research roasted wheat germ (RG) was added in formulation of egg-free milk pudding to enhance its nutritional value and the effects of RG particle size (125, 210 and 354 μm) and quantity (0.0, 2.5, 5.0, 7.5 and 10%) on the quality, nutritional and sensory properties of the resulting pudding were investigated. Reducing the particle size of RG significantly altered its chemical composition but had no significant effect on its antioxidant activity. Increasing the level of RG in the pudding, reduced pH and syneresis while increased dry matter content, hardness, cohesiveness and gumminess of the product. The quantity of RG had more effects on physicochemical properties of the puddings than changing the particle size. Based on the sensory evaluation results, the most acceptable sample was obtained by addition of 7.5% RG with a particle size of 125 μm.

## 1. Introduction

The demand for functional foods with documented health benefits is growing rapidly, since they can offer a unique opportunity to transform our daily foods into affordable health promoting delivery vehicles for the growing population [1,2]. However, most of the available functional foods are marketed as premium products and only affordable by a narrow segment of consumers, which is a major hurdle against employing their full potential. There is also an urgent need to move towards a more sustainable and environmentally friendly food production system to produce highly nutritive foods for all, to enhance public health and address the nutritional demand of the growing number of patients suffering from diet-related diseases including chronic diseases and mineral deficiency [3]. Thus, identifying and using highly nutritive plant-based products are essential to address these issues.

Grains are economical and rich sources of bioactive compounds, proteins and nutrients required for a healthy life, to combat the rapid increase in chronic diseases as well as mineral deficiency, known as hidden hunger around the world. However, grain processing such as milling results in grain loss specially the loss of nutrient dense parts including germ and bran. Therefore, employing practical strategies to reduce grain wastage and valorisation of the by-products are of great importance in fighting food waste and improving the nutritional value of foods at an affordable cost. 

Wheat germ is an economical and nutrient dense by-product of wheat milling factories. In 2012, about 120,000 tons of wheat germ were produced from wheat milling with an anticipated increase in the recent years due to higher wheat production and consumption [3]. Wheat germ offers an excellent choice for food valorisation, product upcycling and circular economy in the food industry. It contains 27–30% protein, 9–17% lipid, 50–51% carbohydrate (mainly simple sugars) and 10–14% dietary fibre [4,5]. Wheat germ is considered as an economical source of high-quality plant proteins which are comparable to egg and milk proteins and contains a high amount of essential amino acids (e.g., lysine) [3,5,6,7]. It is the richest plant source of α-tocopherol (vitamin E) and contains high quantities of vitamin B group and phytochemicals (e.g., ferulic acid and phytosterols) [8]. Wheat germ is considered as a functional food by showing medically proven anticancer, antioxidant, antihyperlipidemia and hypocolesterolemic effects [8,9,10,11,12]. 

Despite the benefits of wheat germ, its applications as a food ingredient are limited and it is mainly wasted as animal feed. It is mostly attributed to its high unsaturated lipids and enzyme activity resulting in rapid rancidity issues including off-flavour, reducing nutritional quality and shelf-life of the wheat germ [4,7,13]. These undesirable effects have been reported for some bakery products enriched with raw wheat germ [14,15]. Our previous research highlighted a significant reduction in sensory attributes of chilled desserts made with raw wheat germ as panellists reported a raw wheaty taste and unpleasant smell requiring further research to fix these issues [16]. 

To overcome wheat germ limitations in food industry, dry heating (roasting) can be applied as an available and practical thermal processing method to inactivate enzymes and reduce moisture content as well as enhancing colour and flavour in wheat germ [5]. Roasting can also destroy the anti-nutrient phytate and eliminate some pathogens in wheat germ [17]. Different combinations of time and temperature have been introduced for roasting wheat germ with minimal damage to the nutrients. It has been documented that roasting wheat germ at 180 °C for 5 to 20 min can successfully enhance its shelf-life, flavour and aroma as well as increasing antioxidant activity [18,19]. Roasted wheat germ has received more attention in the manufacturing of some healthy snacks including biscuits and cookies [20,21].

The enrichment of bakery products such as bread, cake and cookies with wheat germ has been reported [14,15] while scarce knowledge is available on addition of wheat germ to other food groups such as milk-based products. Thus, providing information on the quality and functionality of the milk-based products enriched with wheat germ can encourage food industry to include wheat germ in manufacturing new healthy foods. Accordingly, some new dairy products have been developed using wheat germ including white cheese [22], dairy drinks from sweet whey and butter milk [23], labneh [24], yoghurt-like fermented products [25,26,27] and chilled dairy dessert [16]. 

Milk pudding is a popular product around the world amongst children, elderlies, athletes and health-conscious consumers [28]. Thus, addition of wheat germ to milk pudding may offer a new, healthier and more nutritious type of snack to the customers. 

The major objective of this research was to produce a novel value-added milk pudding using wheat germ to improve the nutritional value of the pudding and also to decrease wheat wastage during milling. Viable strategies (roasting, optimizing the level of addition and particle size control) were applied to combat some of the shortcomings of value addition process including loss of sensory attributes and quality failure. 

## 2. Material and Methods

### 2.1. Materials

Raw wheat germ was supplied by Sepidan Wheat Milling Factory (Sepidan, Fars Province, Iran). Pudding ingredients including gelatine powder was obtained by Behin Azma (Shiraz, Iran), white fine sugar and fresh pasteurized skim milk with 1% fat content were purchased from a local supermarket. Analytical grade chemicals used in this research were obtained from Merck Company (Darmstadt, Germany).

### 2.2. Methods

#### 2.2.1. Roasting of Wheat Germ 

To stabilize wheat germ, a thin layer of raw wheat germ (1 cm thickness) was spread over a perforated aluminium tray and roasted at 180 °C for 15 min in an electrical oven. It was then cooled down to room temperature for 1 h. This method can result in high wheat germ stability [9,18]. The roasted sample was called (RG).

#### 2.2.2. Preparation of RG with Various Particle Sizes

RG was ground once in an Gosonic grinder (Model GCG705, Beijing, China) and sieved through stainless-steel sieves (ASTME: 11, Tehran, Iran) to obtain different particle sizes. Preliminary tests showed that RG can be readily ground to the average particle sizes of 125, 210 and 354 μm while achieving smaller particle sizes was not practical. Ground RG samples were packed in sealed plastic bags and kept at 4 °C. 

#### 2.2.3. Chemical Analysis 

Approved methods of AACC (2000) were used to determine chemical composition of RG (Methods 46–10 for protein, 44–19 for moisture, 08–01 for ash, 30–10 for fat and 32–05 for fiber content) [29]. 

#### 2.2.4. Estimation of Total Phenolic and Flavonoid Content and Antioxidant Activity of RG

Extraction of polyphenols was conducted based on the method using methanol (99.5%) [30]. Aluminium chloride (10%, 0.1 mL), potassium acetate (1 M, 0.1 mL) and distilled water (2.8 mL) were added successively to the extract solutions mixed well. The absorbance of the samples at 415 nm was recorded after 30 min of incubation at ambient temperature. A standard calibration plot was generated using of quercetin with various concentrations less than 100 µg/mL and the flavonoid concentration (mg quercetin equivalent/g of sample) was calculated from the calibration plot [31].

Folin–Ciocalteu’s reagent method was used to determine the total phenolic content [32]. Samples were mixed with 99.5% methanol, and then 0.5 mL of the mixture (20 mg/mL) was mixed with 0.75 mL of freshly diluted Folin–Ciocalteu’s reagent (1:10 with distilled water) and left for 10 min. Na_2_CO_3_ (2%, 0.75 mL) was added to the sample and incubated for 45 min at ambient temperature. Then the absorbance was measured at 765 nm after using a UV/visible spectrophotometer (Unico, Model 2100 pc, Beijing, China). A calibration curve of gallic acid (0, 25, 50 and 100 µg/mL) was obtained, and the total phenolic content (mg gallic acid equivalents per gram of tested dry samples) was measured. Antioxidant activity of the methanol extract was measured based on scavenging activities of the stable 1,1-diphenyl-2-picrylhydrazyl (DPPH) radical.

#### 2.2.5. Pudding Preparation with RG

The ingredients of the pudding were 10% white sugar, 2% gelatin powder, 88% skim milk (cow milk) and different particle sizes (125, 210 and 354 µm) and concentrations (0, 2.5, 5.0, 7.5 and 10% *w*/*w*) of RG. Preliminary tests showed poor sensory properties of the puddings made with more than 10% RG as they became dark and very thick and hence was not tested. To prepare puddings, RG was first hydrated in 1/3 of the milk and stored at 4 °C for 2 h. The remaining milk was heated up to about 50 °C in a sealed glass container and then gelatin powder and sugar were added slowly and blended for 5 min (Brina blender, Model BHB-341, Germany). The hydrated wheat germ in milk was added to the mixture, sealed and batch pasteurized at 72 °C for 30 min. 

To prepare samples for textural analysis, the hot mixtures obtained in the previous step were transferred in Plexiglas tubular moulds (20 mm diameter and 5 mm height) and capped to prevent dryness. The samples were stored at 4 °C for 24 h (fresh samples) and 15 days (stored samples) before texture analysis tests. For other tests including sensory tests, 68 g of the hot mixtures were transferred into plastic cups (100 mL, 6.0 cm diameter), hermetically sealed and refrigerated for 24 h at 4 °C. 

#### 2.2.6. Determination of Dry Matter and pH 

The dry matter was obtained by deducting the moisture content (obtained by oven drying till a constant weight obtained) from 100. 

To measure pH, the pudding was diluted in distilled water 1:2 (*w*/*w*) and the pH was recorded using a pH meter (Model SK-632PH, Metrohm, Bern, Switzerland) after calibration. 

#### 2.2.7. Textural Properties 

The puddings were removed from the refrigerators and immediately tested for texture profile analysis (TPA) using a texture analyser (Texture Analyzer, TA Plus, Stable Microsystems, Surrey, UK) using a 30 kg load cell. Two successive compressions were applied on the samples using a cylindrical probe (40 mm diameter) to compress the samples to 25% of their heights with an interval time of 10 s. The crosshead speed was maintained at 0.25 mm/s. The trigger force (6 g), pretest speed (5 mm/s), test speed (0.25 mm/s) and posttest speed (5 mm/s) were constant for all samples.

The texture profile parameters including hardness, springiness (elasticity), cohesiveness and gumminess were derived from the force–distance curve [33].

#### 2.2.8. Colour Parameters 

Digital colourimetry method was used to puddings colour parameters [34]. Digital images with 300 dots per inch (dpi) resolution, 62% lightness and 62% contrast were taken under the same conditions from the surface of the samples and analysed using Adobe Photoshop CS 5 Software (Adobe Systems Inc., Beijing, China). 

#### 2.2.9. Determination of Syneresis 

To determine syneresis, the samples were removed from the plastic cups (explained in Section 2.2.5) and compressed between two Whatman filter papers (No. 4) of known weights using a 500 g weight for 10 min at ambient temperature. The increasing weight of filter papers was used to express the syneresis [35].

#### 2.2.10. Sensory Evaluation of the Puddings

Different sensory test was performed by 24 in-house panellists (12 males and 12 females, age between 20 and 40 years) using a 5-point hedonic. The sensory attributes were described by a liking score (where 1 = strongly disliked to 5 = strongly liked). The samples were removed from refrigerator, emptied on transparent plastic plates with random codes and presented to the panellists is standard isolated booths. The colour of the samples was evaluated under day light while other sensory attributes under red light [16]. 

#### 2.2.11. Scanning Electron Microscopy

A small piece of each freeze-dried pudding sample was sticked onto an aluminium sample holder and coated with a thin layer of gold metal using an ion sputter coater (Polaron, SC7640, Cambridge, UK) and examined using a scanning electron microscope (5526, Cambridge, UK) with a working distance of 7.5–9.5 mm and at 20.0 kV.

### 2.3. Statistical Analysis 

A completely randomized design was used for analysis of the triplicates data. Average and standard deviation were calculated using Excel 7. Significant differences between the average values obtained for different characteristics of the puddings due to the various concentrations and particle sizes of RG and storage time were evaluated using analysis of variance techniques (ANOVA) and Multiple Ranges Duncan’s test was used to differentiate between the means (*p* < 0.05) using Statistical Analysis System (SAS) software, version 9.3.1. 

## 3. Results and Discussion 

### 3.1. Proximate Analysis and Antioxidant Activity of RG

Table 1 shows the proximate analysis of RG. For different particle sizes, the moisture content ranged between 3.37% and 3.32%, protein content varied from 33.19% to 36.10%, fat content was in the range of 9.37% to 10.70%, ash content was between 4.21% and 4.78% and fibre content was between 3.90% and 8.55%. These values are close to the values reported previously [5,36]. Particle size of wheat germ had a significant effect on its chemical composition and larger particles contained more protein, fat, ash and particularly fibre content. 

The phenolic and flavonoid contents of the RG were 5.06–5.84 mg/g and 5.32–5.54 mg/g, respectively and the antioxidant activity (IC_50_) varied between 54.30 and 54.35. The results revealed that changing the particle size had insignificant consequence on total flavonoid content and IC_50_ of RG; however, reducing the particle size slightly reduced the total phenolic content from 5.84 to 5.06 mg gallic acid/g. This may be due to the heat generated during milling process (used for reducing the particle size of the germ) casing degradation of the phenolic compounds, however, further studies are required to confirm it. 

Compared to the unroasted wheat germ (raw germ) as tested in our previous study [16], the roasted germ had higher protein, fibre and ash content, darker colour and a pleasant nutty flavour but it had lower moisture, fat and antioxidant content (data not given). 

### 3.2. Dry Matter and pH of the Puddings

As shown in Table 2, the control had a dry matter of 21.57% and a pH value of 6.52 was obtained. Addition of wheat germ enhanced the dry matter content as it contains high quantity of solid materials including protein, fibre, fat and ash. Dry matter of the puddings showed a positive correlation with the wheat germ particle size. This can be attributed to the higher dry matter in the wheat germ with larger particles sizes (see Table 1). pH of the samples varied between 6.34 and 6.52 and the samples containing higher levels of wheat germ showed slightly lower pH values. This can be due to the existence of naturally acidic compounds in the wheat germ including fatty acids, glutamic acid and aspartic acid and antioxidants [37] which can lower the pH of the products. Changing the particle size of wheat germ, however, had an insignificant influence on the pH of the puddings. 

### 3.3. Scanning Electron Microscopy

As Figure 1 shows, the control showed some large voids and holes embedded by milk protein and gelatin while the samples containing RG displayed more compact structures with fewer and smaller holes. The dense and compressed structure of wheat germ puddings can have some effects on textural properties, syneresis and sensory attributes of the samples. Similarly, a dense microstructure has been reported for bread enriched with RG and chilled dairy dessert containing raw wheat germ [15,16].

### 3.4. Texture Analysis

Table 3 shows that with increasing the level of RG and storage for 15 days the hardness, cohesiveness and gumminess increased but springiness (elasticity) of the puddings reduced. Changes in the particle size of wheat germ had no significant effect (*p* < 0.05) on the textural parameters of the samples. The micrographs of the samples (Figure 1) also revealed a more solid internal structure of the puddings containing RG which can elucidate the harder and less elastic texture of samples. Protein and carbohydrates of wheat germ can absorb water and interact with pudding components (mainly milk proteins and gelatin) resulting in firmer, more cohesive and less elastic structures. The interactions between wheat germ and milk components as well as water separation were enhanced during storage which can promote hardness and cohesiveness and reduce springiness of the samples. These results are in line with previously reported findings on addition of raw and processed wheat germ to bread and chilled desserts [15,16].

### 3.5. Colour Determination

The colour parameters of the puddings are presented in Table 4. The lightness (L-value) of the puddings decreased significantly while redness (a-value) and yellowness (b-value) increased with addition of the germ concentration and its particle size. The existence of natural yellowish colourants in the wheat germ including carotenoids and xanthophylls as well as dark pigments produced during roasting of wheat germ as a result of Maillard reactions are the main reasons for the observed colour changes of the puddings. In addition, the compact internal structure of the RG pudding as confirmed by SEM microscopy tests and texture analysis results may affect pudding colour by changing light reflection and absorption from the pudding.

### 3.6. Syneresis of the Puddings

Generally, lower syneresis is desirable in puddings as it affects shelf-life, sensory aspects and customer acceptance of the products. Based on the results (Figure 2) the water released from the samples decreased significantly (*p* < 0.05) with addition of RG level, while it remained unaffected by changing the wheat germ particle size. The decrease in syneresis of the RG samples can be related to the high amount of carbohydrates, protein and fibre content of the germ which can form hydrogen bonds with water molecules due to their large number of hydroxyl groups and hinder water separation from the samples. Similar results were reported for chilled dairy dessert containing raw wheat germ [16]. 

### 3.7. Sensory Evaluation

As Table 5 shows, inclusion of RG improved colour, taste and flavour, mouthfeel and general acceptability of the wheat germ added samples compared to the control. The panellists indicated a slight increase in the sweetness of the samples, as the panellists could recognize a pleasant nutty taste for the RG containing puddings. The light brown colour of the RG puddings (Figure 3) was also acceptable for the panellists. It was noted that the samples containing 7.5% RG had the highest scores compared to the other samples. No obvious correlation between the particle size and the sensory parameters of the samples was observed. In our previous study on addition of raw wheat germ to chilled milk dessert, low sensory scores were obtained for the samples particularly in terms of smell and taste as the panellists [16]. However, this issue seems to be resolved by using RG instead of raw germ which resulted in higher sensory scores of the samples containing RG. 

## 4. Conclusions

This research emphasises the feasibility of wheat germ utilization as an economical plant-based functional ingredient in production of milk pudding to enhance its nutritional value and also as a successful upcycling strategy to reduce food waste in wheat supply chain. 

The main effects of changing the particle size of wheat germ were observed on the chemical composition of wheat germ and dry matter and colour of the puddings, while other pudding properties including pH, texture, sensory properties and syneresis were not significantly affected. Large wheat germ particles (i.e., 354 μm) provide higher protein, fat and fibre content but almost similar amounts of flavonoids and antioxidant activity compared to the small size particles (210 µm). This information may be useful in product formulation to adjust the chemical composition of the end product if required. 

The quantity of the added RG was an important factor influencing the milk pudding quality. Wheat germ can be added up to 10% to enhance dry matter, textural hardness, reduce syneresis and improve sensory properties of the puddings. However, pudding colour became caramel brown by addition of wheat germ. If this becomes an issue, addition of natural and approved food colorants such as caramel, cocoa powder or chocolate may be useful. 

Generally, pudding quality was more significantly affected by wheat germ concentration than the size of particles. Addition of 7.5% RG with a particle size of 210 µm resulted in the most acceptable sample. This can add 2.67 g protein, 0.77 g fat, 0.34 g ash, 0.55 g fibre, 40.65 mg/g total phenolic content and 41.92 mg/g total flavonoid content to 100 g of milk pudding and hence improve the nutrition profile and antioxidant content as compared to ordinary milk pudding. 

## Figures and Tables

**Figure 1 foods-11-01815-f001:**
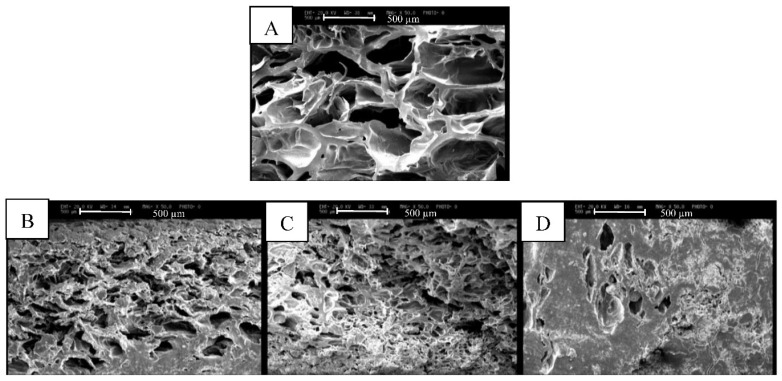
The micrographs of milk pudding containing different levels of roasted wheat germ (RG) with particle size of 210 µm. (**A**): control; (**B**): milk pudding with 2.5% RG; (**C**): milk pudding with 5% RG; (**D**): milk pudding with 7.5% RG.

**Figure 2 foods-11-01815-f002:**
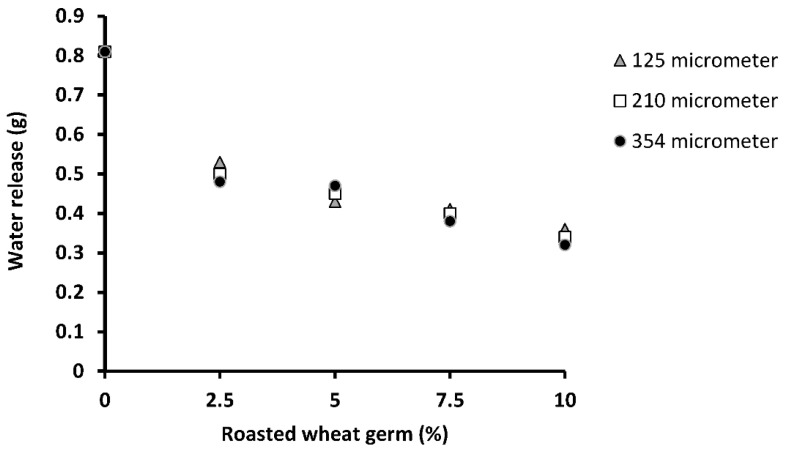
Water release (g) of the milk pudding produced with various concentrations and particle sizes of roasted wheat germ.

**Figure 3 foods-11-01815-f003:**
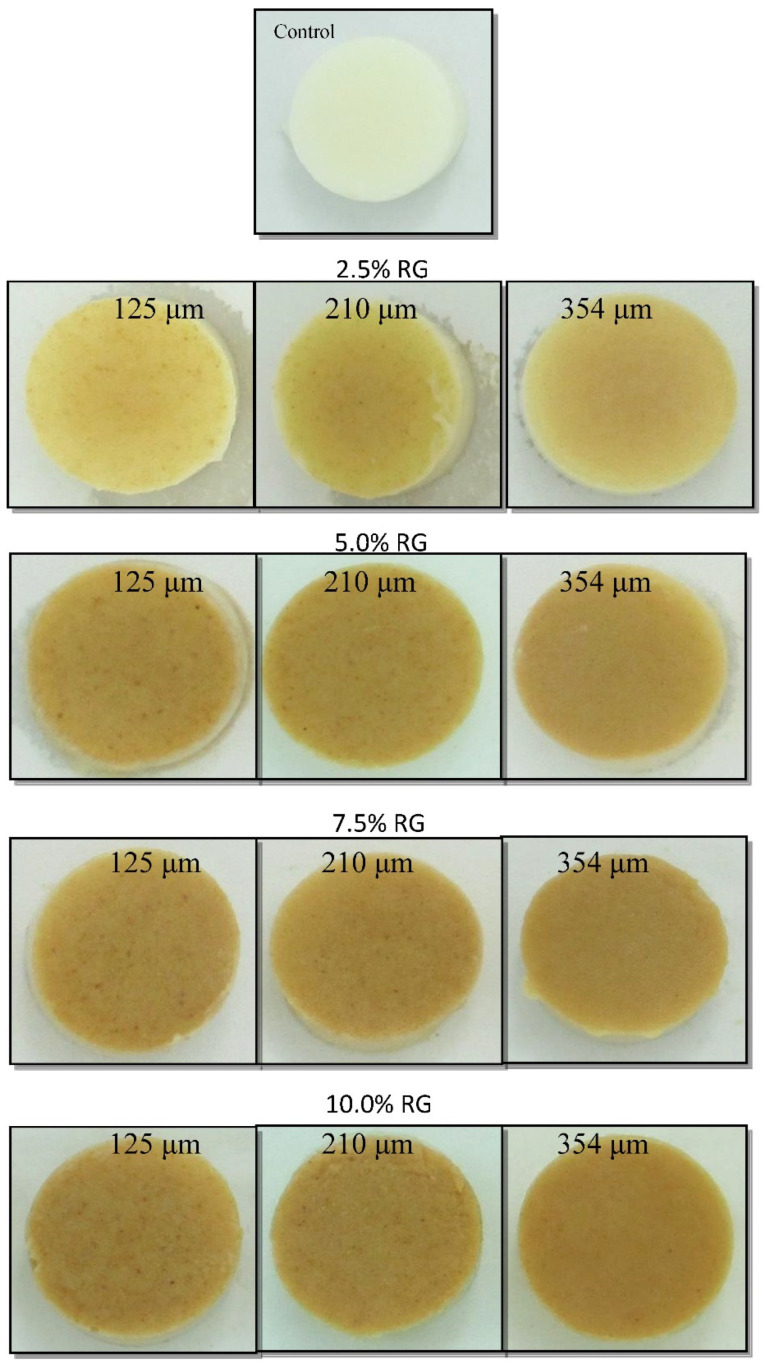
Appearance of milk puddings made with various concentrations and particle sizes of roasted wheat germ (RG). The RG concentration (%) and particle sizes (µm) are shown on the pictures.

**Table 1 foods-11-01815-t001:** Proximate analysis (% dry basis) and antioxidant content and antioxidant activity of roasted wheat germ with various particle sizes.

Particle Size (µm)	Antioxidant Activity (IC_50_ Sample/IC_50_ TBHQ)	Total Flavonoid	Total Phenolic	Fibre	Ash	Fat	Protein ^1^	Moisture
		(mg Quercetin/g Sample)	(mg Gallic Acid/g Sample)					
354	54.35 ^a^	5.32 ^a^	5.84 ^a^	8.55 ^a^	4.78 ^a^	10.70 ^a^	36.10 ^a^	3.37 ^a^
210	54.30 ^a^	5.59 ^a^	5.42 ^ab^	7.32 ^b^	4.56 ^b^	10.23 ^b^	35.70 ^b^	3.36 ^a^
125	54.33 ^a^	5.54 ^a^	5.06 ^b^	3.90 ^c^	4.21 ^c^	9.37 ^c^	33.19 ^c^	3.32 ^b^

Different superscript letters in the same column show significant statistical difference (*p* < 0.05). ^1^ N × 5.45.

**Table 2 foods-11-01815-t002:** Dry matter and pH of milk pudding produced with various concentrations (%) and particle sizes (µm) of roasted wheat germ (RG).

RG (%)	pH			Dry Matter (%)		
	Particle Size (µm)			Particle Size (µm)		
	125	210	354	125	210	354
0	6.52 ^aA^	6.52 ^aA^	6.52 ^aA^	21.57 ^eA^	21.57 ^eA^	21.57 ^eA^
2.5	6.51 ^aA^	6.50 ^aA^	6.47 ^bA^	23.88 ^dC^	24.43 ^dB^	24.90 ^dA^
5	6.39 ^bA^	6.43 ^bA^	6.41 ^bA^	26.07 ^cC^	26.30 ^cB^	26.87 ^cA^
7.5	6.38 ^bcA^	6.34 ^cA^	6.34 ^cA^	28.10 ^bC^	28.37 ^bB^	28.85 ^bA^
10	6.37 ^cA^	6.35 ^cA^	6.35 ^cA^	30.33 ^aC^	30.90 ^aB^	31.87 ^aA^

Different capital superscripts in each row and small superscripts in each column show significant statistical difference (*p* < 0.05).

**Table 3 foods-11-01815-t003:** Texture properties of fresh and stored milk puddings containing different levels of roasted wheat germ (RG) and various particle sizes (µm).

Particle Size (µm)	Fresh Sample (24 h Storage at 4 °C)					Stored Samples (15 Days Storage at 4 °C)				
	RG Level (%)					RG Level (%)				
	0.0	2.5	5.0	7.5	10.0	0.0	2.5	5.0	7.5	10.0
Hardness (g)										
354	53.37 ^aJ^	63.85 ^aH^	73.08 ^aG^	89.61 ^aE^	99.22 ^aC^	57.77 ^aI^	76.29 ^aF^	92.68 ^aD^	106.96 ^aB^	127.09 ^aA^
210	53.37 ^aI^	62.50 ^aG^	72.00 ^aF^	88.77 ^aD^	95.92 ^aC^	57.77 ^aH^	72.23 ^aF^	83.86 ^aE^	101.30 ^aB^	122.83 ^aA^
125	53.37 ^aJ^	61.65 ^aH^	70.50 ^aF^	85.36 ^aD^	95.85 ^aC^	57.77 ^aI^	62.09 ^aG^	78.10 ^aE^	101.28 ^aB^	117.81 ^aA^
Cohesiveness										
354	0.857 ^aE^	0.873 ^aC^	0.876 ^aC^	0.882 ^aB^	0.883 ^aB^	0.857 ^aE^	0.869 ^aD^	0.867 ^aD^	0.885 ^abB^	0.902 ^aA^
210	0.857 ^aE^	0.862 ^aD^	0.862 ^aD^	0.880 ^aB^	0.882 ^aB^	0.857 ^aE^	0.863 ^aD^	0.874 ^aC^	0.887 ^aB^	0.893 ^aA^
125	0.857 ^aD^	0.865 ^aC^	0.878 ^aB^	0.871 ^aB^	0.880 ^aA^	0.857 ^aD^	0.863 ^aC^	0.874 ^aB^	0.877 ^aB^	0.890 ^aA^
Springiness										
354	1.004 ^aA^	0.999 ^aA^	0.995 ^aA^	0.991 ^aAB^	0.981 ^aB^	0.999 ^aA^	0.988 ^aB^	0.986 ^aB^	0.984 ^aB^	0.957 ^bC^
210	1.004 ^aA^	1.001 ^aA^	0.996 ^aA^	0.994 ^aAB^	0.989 ^aB^	0.999 ^aA^	0.996 ^aA^	0.989 ^aB^	0.989 ^aB^	0.972 ^bC^
125	1.004 ^aA^	1.004 ^aA^	0.998 ^aA^	0.997 ^aA^	0.991 ^aB^	0.999 ^aA^	0.997 ^aA^	0.991 ^aB^	0.989 ^aC^	0.988 ^aC^
Gumminess (g)										
354	47.58 ^aJ^	55.82 ^aH^	64.06 ^aG^	79.02 ^aE^	87.63 ^aC^	49.49 ^aI^	66.19 ^aF^	80.29 ^aD^	94.68 ^aB^	114.52 ^aA^
210	47.58 ^aF^	53.87 ^aE^	62.39 ^aD^	78.09 ^aC^	84.67 ^aB^	49.49 ^aF^	62.37 ^aD^	73.55 ^aC^	89.12 ^aB^	109.71 ^aA^
125	47.58 ^aH^	53.37 ^aG^	61.86 ^aF^	74.41 ^aD^	84.36 ^aC^	49.49 ^aH^	53.51 ^aG^	70/02 ^aE^	88.85 ^aB^	106.97 ^aA^

Different capital superscripts in each row and small superscripts in each column show significant statistical difference (*p* < 0.05).

**Table 4 foods-11-01815-t004:** Colour parameters of milk puddings produced with various concentrations (%) and particle sizes (µm) of roasted wheat germ (RG).

RG Level (%)	b-Value			a-Value			L-Value		
	Particle Size (µm)			Particle Size (µm)			Particle Size (µm)		
	125	210	354	125	210	354	125	210	354
0	18.00 ^eA^	18.00 ^dA^	18.00 ^dA^	−9.00 ^dA^	−9.00 ^dA^	−9.00 ^dA^	83. 33 ^aA^	83.33 ^aA^	83.33 ^aA^
2.5	34.33 ^aC^	36.33 ^aB^	37.67 ^aA^	2.00 ^cB^	2.00 ^cB^	2.67 ^cA^	58.67 ^bA^	56.33 ^bB^	55.33 ^bC^
5	31.00 ^bB^	31.68 ^bA^	32.00 ^bA^	2.00 ^cC^	3.33 ^bB^	4.33 ^bA^	48.00 ^cA^	46.00 ^cB^	45.00 ^cC^
7.5	30.00 ^cB^	30.00 ^cB^	31.67 ^bcA^	3.67 ^bB^	4.67 ^aA^	5.00 ^aA^	46.33 ^dA^	45.00 ^dB^	44.67 ^cB^
10	29.33 ^dB^	30.00 ^cA^	30.33 ^cA^	4.33 ^aB^	5.00 ^aA^	5.33 ^aA^	44.33 ^eA^	43.00 ^eB^	42.67 ^dB^

Different capital superscripts in each row and small superscripts in a column show significant statistical difference (*p* < 0.05).

**Table 5 foods-11-01815-t005:** Sensory properties of milk puddings produced with various concentrations (%) and particle sizes (µm) of roasted wheat germ (RG).

RG Level (%)	General Acceptability			MouthFeel			Taste and Flavour			Colour		
	Particle Size (µm)			Particle Size (µm)			Particle Size (µm)			Particle Size (µm)		
	125	210	354	125	210	354	125	210	354	125	210	354
0	4.18 ^cA^	4.18 ^dA^	4.18 ^dA^	3.97 ^cA^	3.97 ^dA^	3.97 ^dA^	4.16 ^cA^	4.16 ^cA^	4.16 ^cA^	4.25 ^bA^	4.25 ^bA^	4.25 ^cA^
2.5	4.33 ^bA^	4.29 ^cA^	4.35 ^cA^	4.39 ^bA^	4.20 ^cB^	4.26 ^cB^	4.20 ^bcAB^	4.28 ^bA^	4.08 ^cB^	4.38 ^aA^	4.25 ^bB^	4.20 ^cB^
5	4.37 ^bA^	4.39 ^bA^	4.37 ^cA^	4.42 ^bA^	4.24 ^cB^	4.38 ^bA^	4.26 ^bB^	4.22 ^bB^	4.41 ^aA^	4.06 ^cB^	4.27 ^bA^	4.17 ^cA^
7.5	4.45 ^aA^	4.53 ^aA^	4.50 ^aA^	4.56 ^aA^	4.48 ^bA^	4.57 ^aA^	4.54 ^aA^	4.62 ^aA^	4.42 ^aB^	4.09 ^cC^	4.58 ^aA^	4.44 ^aB^
10	4.40 ^abA^	4.50 ^aA^	4.43 ^bA^	4.20 ^cB^	4.54 ^aA^	4.31 ^bcB^	4.51 ^aA^	4.12 ^cC^	4.25 ^bB^	4.14 ^cB^	4.07 ^cB^	4.32 ^bA^

Different capital superscripts in each row and small superscripts in a column show significant statistical difference (*p* < 0.05).

## Data Availability

The data are available from the corresponding author.

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
