# Peer review of "Roasted Wheat Germ: A Natural Plant Product in Development of Nutritious Milk Pudding; Physicochemical and Nutritional Properties"

_foods, 2022, doi:10.3390/foods11121815_

Round 1

Reviewer 1 Report

Comments:

This manuscript mainly studied the improvement of nutritional components and textural properties of milk pudding by adding roasted wheat germ. The article compared the particle size and content of roasted wheat germ, and used sensory evaluation and textural properties as indicators to optimize the best addition process. This research extends the applicability of roasted wheat germ, but the following details still need to be revised.

1. Line 105-107: The powder particle size after grinding of the RG sample should be described as a particle size rangeï¼›

2. Line 110-112: In this study, the authors used roasted wheat germ as an additive to improve pudding. What are the obvious changes in the roasting process of wheat?

3. Why does the total flavonoid content increase as the particle size decreases in Table 1?

4. The authors used roasted wheat as a natural additive to improve the nutritional value of the pudding. So what are the special nutritional supplements of roasted wheat?

Author Response

Many thanks for your time reviewing the manuscript and providing valuable feedback. Below, please find the answers to your questions which are also incorporated into the text and highlighted in Yellow.  

  1. Line 105-107: The powder particle size after grinding of the RG sample should be described as a particle size range: In fact, the stated particle sizes are the "average values". The range was not given to make it easy when using particle size in tables and figures. To avoid confusion the word "average" was added to the text. 
  2. Line 110-112: In this study, the authors used roasted wheat germ as an additive to improve pudding. What are the obvious changes in the roasting process of wheat? Thanks for your question. The unroasted wheat germ was studied in our previous research (Majzoobi et al. 2015) so we did not repeat the result here again. Compared to the unroasted wheat germ (raw germ) as tested in our previous study (reference 16), the roasted germ had higher protein, fibre and ash content, darker colour and a pleasant nutty flavour but it had lower moisture, fat and antioxidant content (data not given). This was added to the manuscript. 
  3. Why does the total flavonoid content increase as the particle size decreases in Table 1? In fact, reducing the particle size slightly reduced (not increased) the total phenolic content from 5.84 to 5.06 mg gallic acid/g. This can be due to the heat generation during milling process (used for reducing the particle size of the germ) which needs further studies. This was added to the text. 
  4. The authors used roasted wheat as a natural additive to improve the nutritional value of the pudding. So what are the special nutritional supplements of roasted wheat? A mentioned in the Introduction section, It contains 27-30% protein, 9-17% lipid, 50-51% carbohydrate (mainly simple sugars) and 10-14% dietary fibre [3, 4]. Wheat germ is considered as an economical source of high-quality plant proteins which are comparable to egg and milk proteins and contains high amount of essential amino acids (e.g. lysine) [3, 5-7]. It is the richest plant source of α-tocopherol (vitamin E) and contains high quantities of vitamin B group and phytochemicals (e.g. ferulic acid and phytosterols) [8]. Wheat germ is considered as a functional food by showing medically proven anticancer, antioxidant, antihyperlipidemia and hypocolesterolemic effects [8-12]. More specifically about the roasted wheat germ used in this study, we measured protein, fat, fibre, ash and antioxidants as given in the manuscripts Table 1. Please let me know if further information is required or if the answers need more clarification. 

Reviewer 2 Report

I enjoyed reviewing this article. Although it is a short research paper, it focused on improving popular food. However, there are some minor issues that need to be addressed. Overall, the paper would serve better if the authors could perform a cost analysis and discuss the outcome. 

L(ine) 2: Title. I suggest using: instead of; would be an appropriate choice.

 L33- Add “to” between all and enhance

L64-72- Please add the mechanism/ science of roasting procedure in improving wheat germ to add in food.

L100- You roasted at 150 °C for 15 min in an electrical oven. However, you referred 180 °C for 5 to 20 min roasting earlier (L70). Therefore, it needs to be justified why the experiment carried out such roasting parameters. You may consider adding more literature at L70, different temperature and time combinations.

 L108- What is the science of keeping samples at 4 °C.? How does it help?

L130- …Yen and Chen 1995. This should be in numbers.

L144- …..pasteurized at 72 °C for 30 min. Are you sure? Why did you apply such a longer treatment?

 L209- Please compare your results with literature values.

L220-221- …….had insignificant influence on the pH of the puddings. This is a nice observation you made. However, you need to compare such a trend with literature.

L259-60- What might happen in terms of Maillard reaction if a non-roasted RG was used? What is the impact of Maillard reaction here in your work? In addition, could you please leave some statements on how consumers would respond to the yellowish pudding as we can see in Fig 3?

L290-91- … seems to be resolved…. Please shed more light on this finding. This has a direct relation to the acceptance of the consumers. Why is RG superior to raw germ in your context?

L301- .. economical? Did you do any cost analysis? Would this be feasible in terms of the cost of production?

L313- ….10%.. Is it a common practice? Or your recommendation?

Author Response

Many thanks for your time and also valuable comments. Below please find the answers to the comments. The changes in the text are highlighted in blue. 

L(ine) 2: Title. I suggest using: instead of; would be an appropriate choice. Corrected. 

 L33- Add “to” between all and enhance. Added. 

L64-72- Please add the mechanism/ science of roasting procedure in improving wheat germ to add in food. In the manuscript, it has been stated that: ... practical thermal processing method (roasting) is used to inactivate enzymes and reduce moisture content as well as enhancing colour and flavour in wheat germ [4]. Roasting can also destroy the anti-nutrient phytate and eliminate some pathogens in wheat germ [17]. 

L100- You roasted at 150 °C for 15 min in an electrical oven. However, you referred 180 °C for 5 to 20 min roasting earlier (L70). Therefore, it needs to be justified why the experiment carried out such roasting parameters. You may consider adding more literature at L70, different temperature and time combinations. Thanks for the comment. Actually, it was 180 C which was mistyped in the text. It is corrected now. 

 L108- What is the science of keeping samples at 4 °C.? How does it help? Chilling (storing at 4 C) is a common and well-known storage condition used for Perishable foods to reduce microbial, chemical and enzymatic spoilage 

L130- …Yen and Chen 1995. This should be in numbers. The written reference  (Yen and Chen) was removed as the reference [32] was given previously. 

L144- …..pasteurized at 72 °C for 30 min. Are you sure? Why did you apply such a longer treatment? This is because of the vat pasteurisation equipment we had in the lab. In the food industry or in more advanced lab, HTST is applied. The pasteurisation method we used is working well but in the food industry, they use higher temp and much shorter time using advanced pasteurisation methods. 

 L209- Please compare your results with literature values. No similar results have been quoted previously for reference. 

L220-221- …….had insignificant influence on the pH of the puddings. This is a nice observation you made. However, you need to compare such a trend with literature. No similar results have been quoted previously for reference. 

L259-60- What might happen in terms of Maillard reaction if a non-roasted RG was used? What is the impact of Maillard reaction here in your work? In addition, could you please leave some statements on how consumers would respond to the yellowish pudding as we can see in Fig 3?

Maillard reaction happens at high temperature, so for non-roasted wheat germ there is no Maillard reaction. We have tested non-roasted wheat germ in our previous study mentioned in ref. 16. We found that the raw germ was pale and had a wheaty raw taste and had low stability so it was oxidised rapidly causing off-flavour. Based on these findings we found that roasting can improve taste and flavour and extend the stability of the germ. 

In terms of sensory properties, obviously the highly dark samples were less acceptable which has been discussed in the manuscript. 

L290-91- … seems to be resolved…. Please shed more light on this finding. This has a direct relation to the acceptance of the consumers. Why is RG superior to raw germ in your context? This is based on our previous research (given in reference 16) found that the roasted wheat germ had superior taste, flavour and colour and also improved stability. In the text the differences between the raw and roasted wheat germ are explained (highlighted). 

L301- .. economical? Did you do any cost analysis? Would this be feasible in terms of the cost of production? No cost analysis was done in this research as it was a pilot study and product development research. Many parameters may affect the cost which were out of the scope of this research. However, we anticipate benefits as the wheat germ is cheap and available and the wheat germ dessert can be marketed in the "functional and healthy foods" category many customers are willing to pay a premium for such products. 

L313- ….10%.. Is it a common practice? Or your recommendation? Not a common practice, it varies depending on the formulation, product and process. In this research based on the results, we found that 10% is suitable for pudding. 

Reviewer 3 Report

Mn: Foods-1752589

Roasted wheat germ; A natural plant product in development 2 of nutritious milk pudding; physicochemical and nutritional 3 properties                          Majzoobi et al

The experimental data presented by the authors are pertinent with the journal aims and can be of interest for food industries. However, the submitted version need to be revised.

The originality of presented results as well as the class of consumers to which the investigated formulation of milk pudding could be addressed need to be better emphasised.

The final conclusions (lines 320-323) cannot have a general significance being related to the traits of tested RG. Although, it is correct that addition of RG will increase protein content as well as other compositional traits, the extent of the variation of each trait will depend from the used RG batch.

I find questionable the possibility that panellists have attributed scores with two decimal digits.

Materials and Methods

Have the authors used wheat germ obtained from a single variety or from a mixture of varieties? Have they tested RG from common or durum wheat?

Authors stated the use of skim milk for the test. Which kind of milk was used? From cow, sheep or other? This information is important due to the different characteristics of these milks.

Line 152 – add the duration of oven drying

I suggest to move the Table 1 from the Material and methods into the section Result and discussion.

Results and discussions

Line 214 – value and significance of recorded positive correlation should be included in the text. Is it the only one significant correlation observed?

Line 216 – the pH 6.52 was recorded for the control. In the discussion of results, it is important that authors underline the differences between the range of values relative to pudding containing RG and the value relative to the control.

Tables 2,3,4, 5 – I suggest to revise the format of these Tables. The presented version could generate confusion. I suggest that in all the tables RG % might be reported in a column (as in Table 2 and 5) or in a row (as in Tables 3 and 4).

Fig 2. The scale of y-axis need to be modify to avoid the overlap of experimental values. The submitted version does not allow the correct visualization of results.

Minor remark

Line 36 – the reference 5 is inappropriate and included in the text before the ref 3 and 4.

Line 46 –  A reference relative to the world production should be added.

Line 130 – the reference should be cited as number

Line 206 – the lowest value is 54.30 not 54.33

Author Response

Many thanks for reviewing the manuscript and providing your valuable comments. Below please find the answers to your comments. Changes in the text are highlighted in green. 

The experimental data presented by the authors are pertinent with the journal aims and can be of interest for food industries. However, the submitted version need to be revised.

The originality of presented results as well as the class of consumers to which the investigated formulation of milk pudding could be addressed need to be better emphasised. This has been given in the text as below: 

It has been indicated that the enrichment of bakery products such as bread, cakes and cookies with wheat germ [14, 15] while scarce knowledge is available on addition of wheat germ to other food groups such as milk-based products. Thus, providing information on the quality and functionality of the milk-based products enriched with wheat germ can encourage food industry to include wheat germ in manufacturing new healthy foods. Accordingly, some new dairy products have been developed using wheat germ including white cheese [22], dairy drinks from sweet whey and butter milk [23], labneh [24], yoghurt-like fermented products [25-27] and chilled dairy dessert [16].

Milk pudding is a popular product around the world amongst children, elderlies, athletes and health-conscious consumers [28]. Thus, addition of wheat germ to milk pudding may offer a new, healthier and more nutritious type of snack to the customers.

The final conclusions (lines 320-323) cannot have a general significance being related to the traits of tested RG. Although, it is correct that addition of RG will increase protein content as well as other compositional traits, the extent of the variation of each trait will depend from the used RG batch. Generally, wheat germ is a nutrient-dense component and can fortify foods with protein, fibre, minerals and bioactive compounds. Thus, even if different batches are used and tested similar trend is expected. 

I find questionable the possibility that panellists have attributed scores with two decimal digits. The scores are the "average values" of replicates (not the raw scores from panelists) that is why they have decimal numbers. 

Materials and Methods

Have the authors used wheat germ obtained from a single variety or from a mixture of varieties? Have they tested RG from common or durum wheat? The wheat germ was obtained from a wheat milling factory that uses mixed varieties. It is a common practice to mix wheat varieties in the mills to produce desirable flour. Thus, testing a single variety is not feasible since the available commercial germ comes from a mixture of varieties. Also, the same type of wheat germ was used in all trials and hence the effect of variety is eliminated. 

Authors stated the use of skim milk for the test. Which kind of milk was used? From cow, sheep or other? This information is important due to the different characteristics of these milks. The most common type of milk in the market is cow milk (added in the text). However, testing the type of milk was not the aim of this research. 

Line 152 – add the duration of oven drying. The time varies depending on the initial weight of the sample and based on the Standard Method, the drying continues until a constant weight of the sample is obtained (this was added in the text). 

I suggest to move the Table 1 from the Material and methods into the section Result and discussion. In the version that I am checking it is in the Results and Discussion section but I will make sure it is in the right place. 

Results and discussions

Line 214 – value and significance of recorded positive correlation should be included in the text. Is it the only one significant correlation observed? All the actual values are given in the relevant table and we avoid repeating all of them in the text. Hope I answer your question. 

Line 216 – the pH 6.52 was recorded for the control. In the discussion of results, it is important that authors underline the differences between the range of values relative to pudding containing RG and the value relative to the control.

The pH values for other samples are given as a range to make it easier to read the text. However, all actual data are presented in the tables for further information. Please see below:

The pH of the samples varied between 6.34 and 6.52 and the samples containing higher levels of wheat germ showed slightly lower pH values. 

Tables 2,3,4, 5 – I suggest to revise the format of these Tables. The presented version could generate confusion. I suggest that in all the tables RG % might be reported in a column (as in Table 2 and 5) or in a row (as in Tables 3 and 4).

Thanks for your suggestion, The authors decided to leave the table format as it is because even if we put wheat germ in columns, then we need to put the particle size in the rows and that makes no big improvement. Other reviewers had no comments on this and we have used the same format in our previous publications with no issues.

Fig 2. The scale of y-axis need to be modify to avoid the overlap of experimental values. The submitted version does not allow the correct visualization of results.

On the version I am checking they are correct and I make sure the high-quality version is submitted. 

Minor remark

Line 36 – the reference 5 is inappropriate and included in the text before ref 3 and 4. Thanks, this was corrected in the text and the reference list. 

Line 46 –  A reference relative to the world production should be added. This was added. 

Line 130 – the reference should be cited as number. Corrected. 

Line 206 – the lowest value is 54.30 not 54.33. Corrected.